# An optimized electrotransformation protocol for *Lactobacillus jensenii*

**Elsa Fristot[1], Thomas Bessede[1], Miguel Camacho Rufino[1], Pauline Mayonove[1], Hung-Ju Chang[1], Ana Zuniga[1], Anne-Laure Michon[2], Sylvain Godreuil[3], Jérôme Bonnet[1]\*, Guillaume Cambray[1,2]\***

**1** Centre de Biologie Structurale (CBS), University of Montpellier, INSERM U 1054, CNRS UMR 5048, Montpellier, France, **2** Diversité des Génomes et Interactions Microorganismes Insectes (DGIMI), University of Montpellier, INRAE UMR1333, Montpellier, France, **3** Service de Bactériologie, Hôpital Arnaud de Villeneuve—CHU de Montpellier, Montpellier, France

\* cambray.guillaume@gmail.com (GC); jerome.bonnet@inserm.fr (JB)

**Data Availability Statement:** All relevant data are within the paper and its Supporting Information files.

**Funding:** JB acknowledges support from the INSERM Atip-Avenir program (https://www.

## Abstract

Engineered bacteria are promising candidates for *in situ* detection and treatment of diseases. The female uro-genital tract presents several pathologies, such as sexually transmitted diseases or genital cancer, that could benefit from such technology. While bacteria from the gut microbiome are increasingly engineered, the use of chassis isolated from the female uro-genital resident flora has been limited. A major hurdle to implement the experimental throughput required for efficient engineering in these non-model bacteria is their low transformability. Here we report an optimized electrotransformation protocol for *Lactobacillus jensenii*, one the most widespread species across vaginal microflora. Starting from classical conditions, we optimized buffers, electric field parameters, cuvette type and DNA quantity to achieve an 80-fold improvement in transformation efficiency, with up to $3.5 \cdot 10^3$ CFUs/µg of DNA in *L. jensenii ATCC 25258*. We also identify several plasmids that are maintained and support reporter gene expression in *L. jensenii*. Finally, we demonstrate that our protocol provides increased transformability in three independent clinical isolates of *L. jensenii*. This work will facilitate the genetic engineering of *L. jensenii* and enable its use for addressing challenges in gynecological healthcare.

## Introduction

The vaginal mucosa defines a rich and complex ecosystem that comprises millions of microorganisms collectively referred to as the Döderlein flora [1]. The healthy vaginal flora is mainly composed of lactobacilli (ca. 95% of the population). These bacteria have been shown to play important roles in preventing dysbiosis in favor of other anaerobic pathogenic bacteria, fungi and viruses, which are otherwise present at low abundance. In particular, lactic acid, hydrogen peroxide and other antimicrobial substances produced by these lactobacilli have been shown to inhibit the growth of pathogenic bacteria in the vagina and reduce sexually transmitted infections [2, 3]. Lactobacilli belong to the order of Lactobacillales, which regroups an ensemble of functionally related but phylogenetically different bacteria [4]. They are Gram-positive, facultative anaerobic, non-spore-forming bacteria. Their capacity to ferment a wide variety of

inserm.fr/en/about-us/atip-avenir-program) and the Bettencourt-Schueller Foundation (https:// www.fondationbs.org). GC acknowledges support from the CNRS Atip-Avenir program (https://www. insb.cnrs.fr/fr/atip-avenir). The CBS acknowledges support from the French Infrastructure for Integrated Structural Biology (FRISBI, https://frisbi. eu/; ANR-10-INSB-05-01). The funders had no role in study design, data collection and analysis, decision to publish, or preparation of the manuscript.

**Competing interests:** The authors have declared that no competing interests exist.

sugars into lactic acid is at the root of wide-ranging adaptation to specific niches. Commensal lactobacilli species are particularly abundant on nutrient-rich mucous membranes and in food. These bacteria are thus highly prevalent within various human microbiomes, and several have been shown to contribute to the body's defense against pathogenic microorganisms [3].

Broadly speaking, lactobacilli and other lactic acid bacteria hold great promises for various biotechnological and therapeutic applications (**Fig 1A**) [5, 6]. Flagship examples include the development of probiotic treatments [7] and *in situ* production of mucosal vaccines [8]. Although providing an effective way to widen industrial applicability, the genetic engineering of these highly diverse lactobacilli species remains limited by notorious difficulties in developing efficient biotransformation procedures [6, 9–11]. To a large extent, this low transformability is linked to the thick peptidoglycan layer that covers their Gram-positive membrane. As the membrane composition and properties can vary substantially between species, transformation protocols must be optimized on an *ad hoc* basis.

Originally described in 1969 by Gasser *et al.* [12], *Lactobacillus jensenii* is prevalent in the majority of vaginal microbiomes characterized across a diverse sample of ethnic groups worldwide [1]. In most of these groups, *L. jensenii* consistently represents ca. 20% of the vaginal flora. As such, *L. jensenii* would constitute a particularly suitable chassis for engineering the vaginal microbiota. However, functional knowledge regarding this species is particularly sparse. To the best of our knowledge, only two research groups have worked on *L. jensenii* engineering to date: Osel Incorporation has engineered *a strain* to secrete therapeutic effectors capable of reducing HIV infection [13] and Hervana.bio [14] engineered a strain to secrete contraceptive molecules that recently entered clinical trials. While these achievements readily demonstrate the strong potential of this bacterium, most of the tools and techniques developed in these private ventures have not been made available to the wider scientific community, thus limiting its use for further biotechnological innovations.

In this work, we aimed at developing a transformation protocol sufficiently efficient to support the construction of medium-sized genetic libraries (*i.e.* at least $10^3$ CFUs/μg DNA)in *Lactobacillus jensenii* (**Fig 1B**). Electroporation is the only effective method for the transformation of Gram-positive bacteria in general, and lactobacilli in particular [9–11]. Here, we started from protocols widely used for related *Lactobacillus* species [15, 16] and optimized several key parameters related to competent cell preparation and electroporation procedure (**Fig 1C**). We defined a transformation protocol that can yield up to $3.5 \cdot 10^3$ CFUs per microgram of plasmid DNA and is effective on multiple strains of *L. jensenii*. We further identified 2 unrelated plasmids that can be maintained in *L. jensenii* and used to drive expression of a reporter gene.

## Material and methods

### Strains and media

The reference strain of *Lactobacillus jensenii* used in this study is a vaginal isolate *(Gasser et al.; ATCC 25258)*. Three other independent clinical isolates (Strain#1 ID: 99220272882; Strain#2 ID: 99220273430; Strain #3 ID: 99220254921) of *Lactobacillus jensenii* were obtained from hospital bioresource bank from patient suffering vaginal dysbiosis (Bacteriology lab, Arnaud de Villeneuve University hospital, Montpellier, France). Cells were grown in liquid MRS medium (De Man, Rogosa and Sharpe, DIFCO) at 37˚C in anaerobic conditions using GasPak anaerobic jar (BD GasPak™ EZ container systems), without agitation. MRS composition is as follow: peptone proteose: 10 g/L, beef extract: 10 g/L, yeast extract: 5.0 g/L, dextrose: 20 g/L, polysorbate 80: 1.0 g/L, ammonium citrate: 2.0 g/L, sodium acetate: 5.0 g/L, magnesium sulfate: 0.1 g/ L, manganese sulfate: 0.05 g/L, dipotassium phosphate: 2.0 g/L. When appropriate, erythromycin (E6376 Sigma-Aldrich) was added to the media at 0.5 μg/mL to select for transformants.

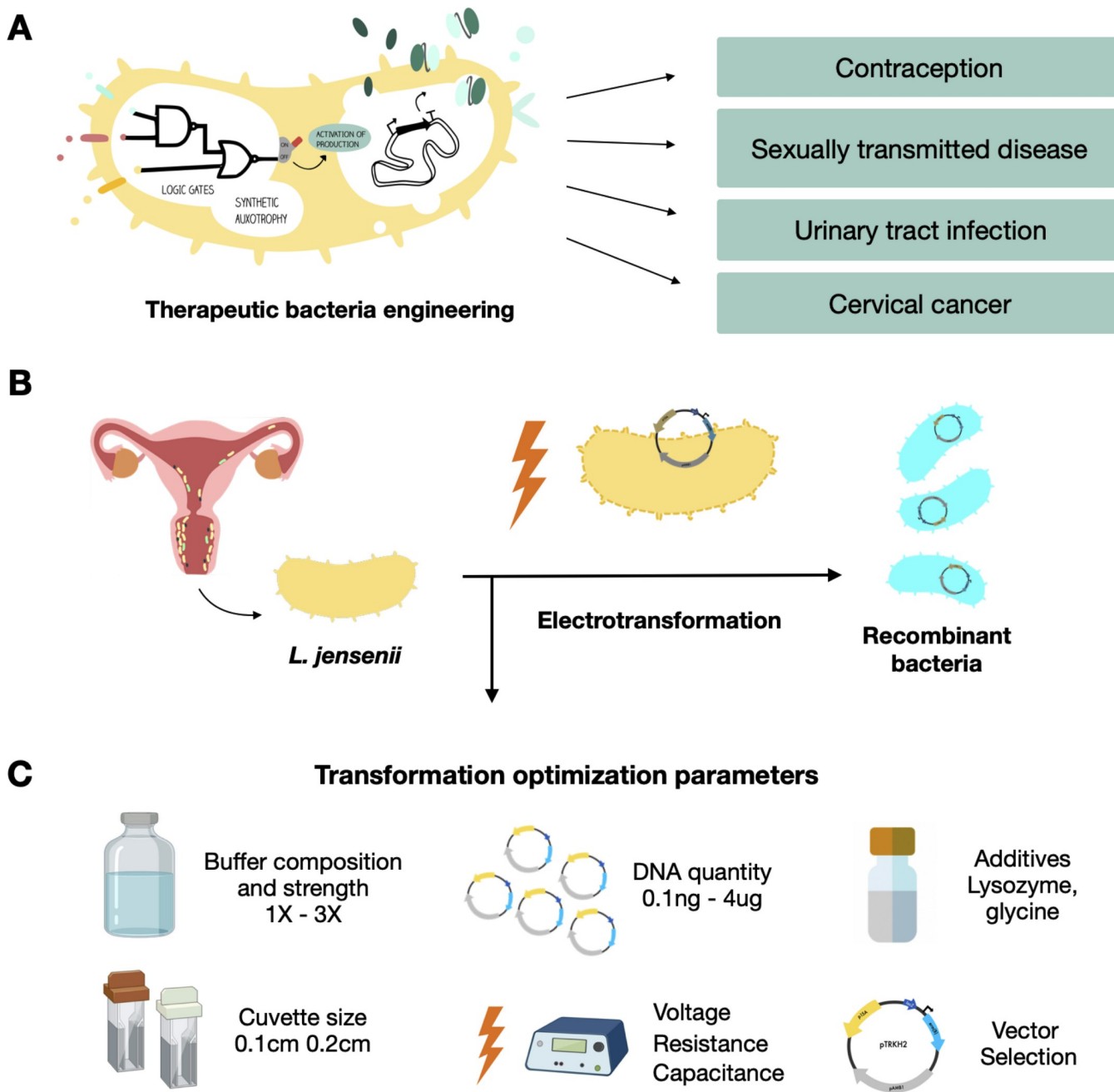

**Fig 1. Potential applications to vaginal microbiota engineering and electroporation optimization. A.** Potential applications of engineering the vaginal microbiota.**B.** *L. jensenii* is a candidate to develop genetically engineered probiotics for the vaginal microflora. Electrotransformation is the most effective method to transform DNA into *Lactobacillus spp*. **C.** Parameters described as important to optimize electroporation in lactic acid bacteria that were tested in this study.

### *Lactobacillus jensenii* ATCC 25258 erythromycin susceptibility test

*L. jensenii ATCC 25258* susceptibility to erythromycin was determined by growing cells on MRS agar plate and MRS liquid medium supplemented with a range of antibiotic concentrations (0.5 μg/mL, 1.0 μg/mL, 2.5 μg/mL, 5.0 μg/mL, 7.5 μg/mL, and 10.0 μg/mL). Cultures were incubated for two days at 37°C under anaerobic conditions in a GasPak jar before visual evaluation of growth.

## Plasmid vectors

We selected 3 shuttle vector plasmids described in the literature as being widely portable among lactobacilli. pLEM415-ldhl-mRFP11 (a gift from Sujin Bao, Addgene plasmid # 99842) is a pBLUESCRIPT derivative carrying an unnamed lactobacilli origin of replication, a high-copy version of the *E. coli* colE1 origin of replication, an erythromycin resistance gene and a *mRFP1* expression cassette. pTRKH2 (kind gift from Rodolphe Barrangou & Todd Klaenhammer, Addgene plasmid #71312) is a theta replicative plasmid carrying a pAMβ1 origin, a p15A *E. coli* replication origin and an erythromycin resistance gene *Erm(B)*. pTRK892 (kind gift from Rodolphe Barrangou & Todd Klaenhammer, Addgene plasmid #71803) is a high copy shuttle vector bearing a pWVO1 replication origin that is functional in both gram negative and positive species. pTRK892 also carries a *GusA* reporter cassette under *ppgm* promoter and erythromycin *Erm(C)* resistance gene.

## Competent cells preparation

Optimized preparation of competent cells from *L. jensenii ATCC 25258* and the three clinical isolates is performed as follows. Cells are grown overnight in 50 mL MRS at 37˚C without agitation. 98 mL of fresh MRS medium is then inoculated with 2 mL of the overnight culture and incubated at 37˚C without shaking to an OD of 0.5–0.6 (ca. 4–5 hours) in a GasPak anaerobic jar (BD GasPak™ EZ container systems). Cells are harvested a first time by centrifugation at 4,000 g for 5 min. From then on, cells are always maintained on ice or at 4˚C in refrigerated centrifuges. Supernatant is discarded and pelleted cells resuspended in 15 mL ice-cold 3X SMEB electroporation buffer (1X Sucrose Magnesium Electroporation Buffer: 298 mM sucrose, 1 mM MgCl2 in cold sterile water), and centrifuged at 4,500 g for 7 min. This procedure is repeated twice. Cell pellets are then concentrated in 1.0 mL of ice-cold 3X SMEB buffer and 200 µL aliquots can be placed at -80˚C for long term storage or alternatively used directly for electroporation up to one hour after preparation.

## Electroporation

A GenePulser X Cell apparatus (Bio-rad Laboratories, Richmond, CA) was used for all electroporation experiments. The final optimized electroporation procedure is as follows: 200 µL competent cells are thawed on ice, transferred in a 0.2 cm electroporation cuvette (Bio-rad Laboratories, Richmond, CA) after addition and gentle mixing of 1 µg of DNA, and incubated on ice for 10 minutes. Cells are electroporated after drying the electrodes with a clean paper wipe using the following parameters: 12.5 kV/cm, 400 Ohm and 25 µF capacitance. Efficient electroporation events show a time constant in the 8.0–10.5 ms range. Directly after the electric pulse, cells are rapidly and gently resuspended in 800 µL antibiotic-free MRS medium preheated at 37˚C and transferred to culture tubes for a 3 hours incubation period at 37˚C without shaking in GasPak containers. 100 µL of the resulting culture is then spread on MRS agar plate (1% agar) supplemented with 0.5 µg/mL erythromycin using glass beads (0.4 cm diameter). As necessary, all cells can also be concentrated by centrifugation at 4,000 g for 5 min and gently resuspended in 100 µL MRS before spreading.

Transformation efficiencies were calculated after counting colonies upon 48 hours of incubation at 37˚C in GasPak containers. Effective transformation of *L. jensenii* was confirmed by liquid culture-based PCR targeted to the 16S RNA gene and to a segment of the plasmid pTRKH2, as well as by beta-glucuronidase assays for plasmid pTRK892. Primers sequences are listed in S1 Table.

### Liquid culture-based PCR

Colonies were inoculated in 1 mL MRS supplemented with 0.5 μg erythromycin and grown for 24h at 37˚C in a GasPak jar without shaking. Cultures were centrifuged at 5,000 g for 5 min and the pellet resuspended in 100 μL PBS. One microliter of this concentrated culture was then used to perform verification colony PCR using the KAPA polymerase (Biosystem), following the vendor's instructions.

### Beta-glucuronidase reporter assays

pTRK892 transformation was confirmed by assaying the activity of the beta-glucuronidase *GusA*. Colonies obtained upon electrotransformation of pTRK892 were used to inoculate 1 mL MRS culture supplemented with 0.5 μg erythromycin. Cultures were grown for 48h at 37˚C in a GasPak jar without shaking and centrifuged at 6,000 g for 5 minutes. Supernatants were discarded and pellets were resuspended in 400 μL resuspension buffer (50 mM NaH2PO4). Samples were then incubated for 30 minutes at 37˚C after addition of 25 μL permeabilization solution (9:1 acetone to toluene v/v). Beta-glucuronidase activity was evaluated qualitatively by visual observation of a green to blue coloration of the samples 30 minutes after addition of 5 μL of X-gluc stock solution (10 mg/mL in DMSO; Sigma Aldrich).

## Results

### Choice of plasmids and initial transformation conditions

Accurate measurements of transformation efficiency rely on the maintenance of the transformed plasmids by recipient cells, as well as the expression of phenotypes that can be used to select and/or identify transformants. As most functional genetics of *L. jensenii* was performed in the context of private research endeavors, little information regarding which plasmids might be suitable for this species is publicly available. We thus selected three vectors that bear replication origins that are widely functional in other lactobacilli species and that are readily available on Addgene: pTRKH2, pTRK892, and pLEM415-ldhl-mRFP1. All these plasmids comprise an expression cassette conferring resistance to erythromycin (see **Table 1**, Material and Methods and **S1 Fig**).

We started by testing two widely used—but largely different—transformation protocols for lactobacilli initially reported by Berthier *et al.* [15] and Luchansky *et al.* [22] using the vector pTRKH2 (Material and Methods). After growing electroporated bacteria on MRS agar plates and liquid cultures tubes supplemented with increasing concentrations of erythromycin, we determined that a working concentration of 0.5 μg/mL enabled unambiguous selection of transformant cells in MRS. In these conditions, the Luchansky protocol resulted in significantly more transformants, though the absolute number of colonies obtained remained

**Table 1. Designation, characteristics and transformation frequency of plasmids tested in *Lactobacillus jensenii* ATCC 25258.**

| Plasmid | Resistance | Copy number | Size | Origin | Source | Ref | Average transformation frequency | |
|---|---|---|---|---|---|---|---|---|
| | | | | | | | before optimization | After optimization |
| pTRKH2 | *Erm B* | High copy | 6,719 | pAMβ1 | Addgene | 21 | $4.7 \times 10^1$ | $2.0 \times 10^3$ |
| pTRK892 | *Erm C* | High copy (gram -) Low copy (gram +) | 6,470 | pWV01 | Addgene | 22, 23 | $3.2 \times 10^1$ | $3.5 \times 10^3$ |
| pLEM415-ldhl-mRFP1 | *Erm* | High copy | 7,357 | *L. fermentum* | Addgene | 24, 25 | $1.2 \times 10^1$ | $3.3 \times 10^1$ |

Abbreviation: Erm, erythromycin 0.5 μg/mL in *Lactobacillus jensenii*. Transformation efficiency are expressed as erythromycin resistant colonies forming unit per μg of plasmid (CFUs/μg).

relatively low ($4.7 \cdot 10^1$ CFUs/μg plasmid on average; **Fig 2A**). We next attempted electroporation of the two other vectors, but could only obtain transformants with pTRK892, even under lower antibiotic pressure (**Table 1**). This prompted us to use the Luchansky protocol with plasmid pTRKH2 as a starting point to optimize the transformation of *L. jensenii* ATCC 25258.

### Protocol optimization for improved *L. jensenii* transformation

Using a greedy hierarchical optimization strategy, we successively varied several parameters previously reported to impact electrotransformation of *Lactobacilli* in different sources [11, 15–17]: (i) Buffer concentration, (ii) Glycine concentration, (iii) cuvette length, and (iv) strength of the electric field (**Fig 1C**).

Using 0.1 cm electroporation cuvettes and electric field parameters as described in Luchansky et al. [16], we first varied the concentration of the SMEB buffer between 1 and 3X. We found that transformation efficiency increased with concentration, yielding up to $4.8 \cdot 10^2$ CFUs/μg on average with the 3X buffer (an 8-fold improvement; **Fig 2B**). Higher concentrations led to the formation of electric arcs that killed the cells, producing no transformants.

We next investigated the addition of glycine, which is known to weaken the bacterial cell wall [17]. Using a 3X SMEB concentration, we tested different glycine weight/volume ratio (0.0%, 0.5%, 1.0%, 1.5% and 2.0%) and observed a non-monotonic impact on transformation efficiency with a maximum average of $7.5 \cdot 10^2$ CFUs/μg obtained with 2.0% glycine (**Fig 2C**). Higher glycine titers led to markedly reduced growth rate in liquid culture (and eventually to cell death for glycine titers over 5.0%) [17].

We next replaced the 0.1 cm cuvettes by 0.2 cm cuvettes (**Fig 3A**). This improved the average transformation efficiency to $1.4 \cdot 10^3$ CFUs/μg of plasmid on average. This 2.5-fold improvement over 0.1 cm cuvettes strikingly mirrors the increased quantity of cells loaded into the cuvette (from 80 to 200 μL of competent cells).

We then measured the impact of using increasing quantities of plasmid DNA, from 0.5 μg to 4.0 μg DNA per transformation (**Fig 3B**). While we observed a spectacular improvement

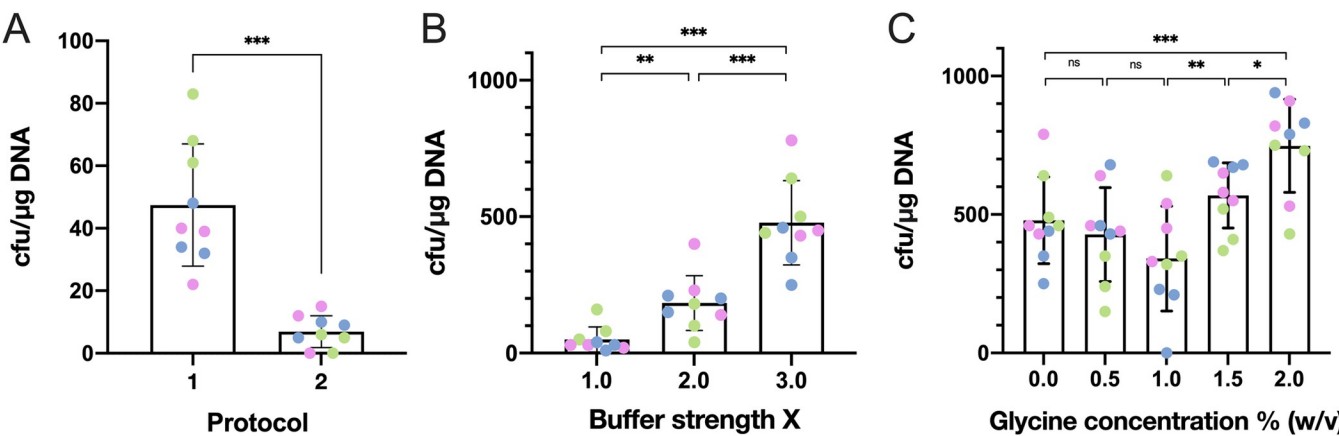

**Fig 2. Optimization of competent cells preparation for *L. jensenii* ATCC25258 using pTRKH2. A.** Comparison of widely used electroporation protocols for *Lactobacillus spp*. Shown are transformation efficiency of *Lactobacillus jensenii* observed using original protocols described by Luchansky et al. (protocol 1) [22] and Berthier et al. (protocol 2) [21]. **B.** Increasing buffer concentration improves transformation efficiency. Shown are transformation efficiency of electrocompetent *L. jensenii* prepared at 1, 2 or 3X the concentration described in Luchansky et al. [16]. More concentrated buffers led to arcing. **C.** Addition of glycine improves transformability. Shown are transformation efficiency of electrocompetent *L. jensenii* prepared in 3X buffer with increasing titers of glycine (expressed as weight to volume percentage). Titers higher than 2.0% lead to markedly reduced growth rate. All electroporation were performed with 1 μg pTRKH2 in 0.1 cm electroporation cuvettes and electric parameters as follows: voltage: 6.5 kV/cm, resistance: 600 Ohm, capacitance: 25 μF. 3 biological replicates of 3 technical replicates were performed on different days (data are shown in S2 Table). Technical replicates are shown in the same color. (Wilcoxon signed-rank test p-value, ns $p > 0.05$, * $p < = 0.05$, ** $p < = 0.01$, *** $p < = 0.001$ and **** $p < = 0.0001$).

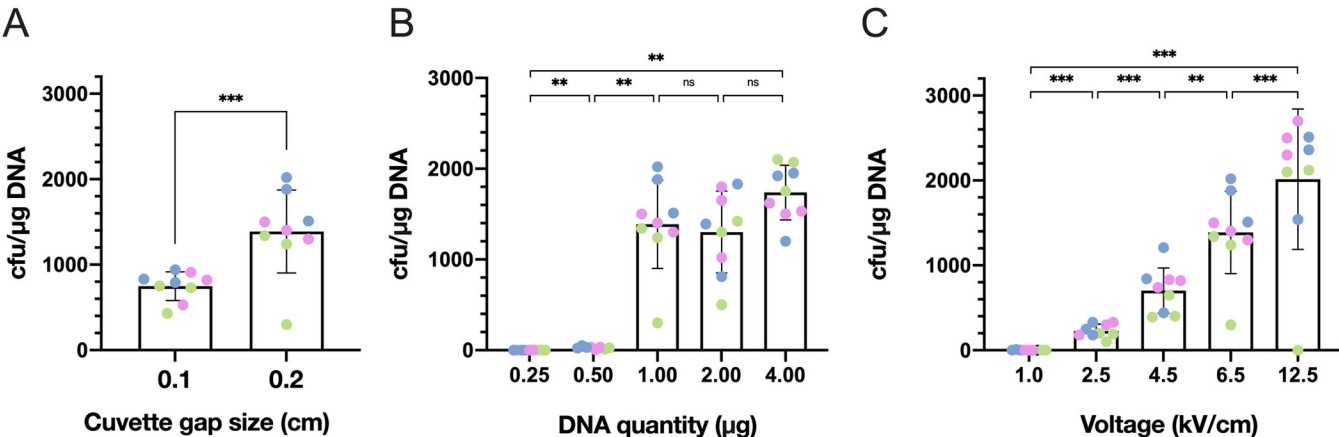

**Fig 3. Optimization of electroporation procedure for *L. jensenii 25258* using pTRKH2 vector. A.** Increasing electroporation cuvette size yields a 2.5-fold increase in transformation efficiency. Shown are transformation efficiency of electrocompetent *L. jensenii* in 0.1 cm or 0.2 cm cuvette gap size. **B.** Increasing DNA quantity impacts transformation efficiency in a step-like manner. Shown are transformation efficiency of electrocompetent *L. jensenii* electroporated in 0.2 cm cuvettes with different DNA quantities, as indicated. Quantities above 4 μg (or volume above 5 μL) frequently result in electric arcs. **C.** Increasing electric pulses' voltage improves transformation efficiency. Shown are transformation efficiencies for electrocompetent *L. jensenii* electroporated in 0.2 cm cuvettes with 1 μg DNA at different voltage intensities, as indicated. Increasing the voltage above 12.5 KV/cm causes arcing and kills the cells. All electroporations were carried with vector pTRKH2, 3X SMEB, 2% glycine, 6.5 kV/cm voltage (unless otherwise indicated), 400 Ω resistance, 25 μF capacitance. 3 biological replicates of 3 technical replicates were performed on different days (data are shown in S2 Table). Technical replicates are shown in the same color. (Wilcoxon signed-rank test p-value, ns p > 0.05, $^{*}$ p <= 0.05, $^{**}$ p <= 0.01, $^{***}$ p <= 0.001 and $^{****}$ p <= 0.0001).

from 0.5 μg to 1.0 μg, increasing DNA quantities over this value did not yield major improvements, but increased the likelihood of electric arc formation. Finally, we tested the impact of varying the strength of the electric field from 1.0 to 12.5 kV/cm using 1 μg DNA, and found that increasing the pulse's voltage improved transformation efficiency (**Fig 3C**).

Our final transformation procedure, which consists in using 3X SMEB buffer, 1 μg of DNA, 2% glycine, 0.2 cm cuvette with 12.5 kV/cm voltage, yields a transformation efficiency close to $2 \cdot 10^3$ CFUs/μg of plasmid, exceeding our target efficiency of $10^3$ with plasmid pTRKH2. We then proceeded to re-test the transformation of the 2 other vectors: pTRK892 led to slightly better results than pTRKH2, while pLEM415 yielded a 100-fold lower transformation efficiency, indicating the origin of replication as an important choice for better transformation efficiency (**Fig 4A**). We thus obtained with pTRK892 a maximum transformation efficiency of $3.5 \cdot 10^3$ CFUs/μg plasmid. The final protocol is fully detailed in S1 Protocol.

To verify that the colonies selected upon transformation corresponded to *L. jensenii ATCC 25258* bearing a plasmid, we performed PCRs targeted to the 16S RNA gene and to a segment of pTRKH2 on 16 randomly picked colonies (see Material and Methods). All putative transformants yielded amplicons of the expected sizes. We also used a beta-glucuronidase assay on isolates transformed with pTRK892. The production of a blue color from the processing of substrate X-Gluc confirmed the successful expression of the *gusA* reporter gene by the transformed cells (**Fig 4B**). Finally, in an effort to assess whether our optimized protocol is generally effective in other *L. jensenii* isolates, we subjected three independent clinical isolates of *L. jensenii* to both the original and optimized electroporation procedures. Although a lower absolute number of transformants were obtained as compared to the reference strain, our optimized protocol afforded a 28-fold increase in transformation efficiency on average across these isolates (**Fig 4C**). These data suggest that our protocol can be of general use for engineering clinical isolates of *L. jensenii*.

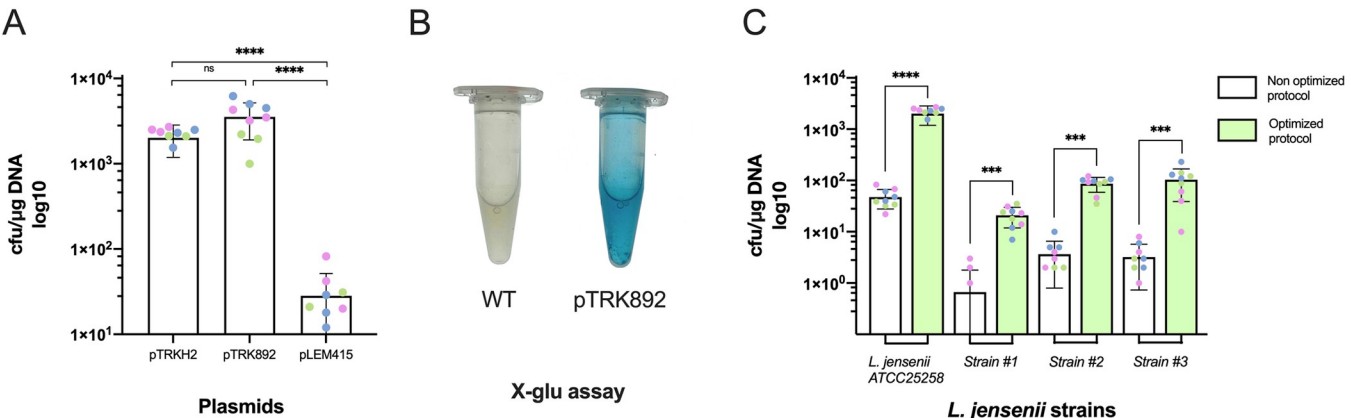

**Fig 4. Optimized transformations of various vectors and unrelated *L. jensenii* strains. A.** Transformation efficiency of different plasmids using the optimized protocol. Shown are transformation efficiencies obtained for plasmids pTRKH2, pTRK892 and pLEM415-ldhl-mRFP1. While pTRKH2 and pTRK892 yield similarly high efficiencies, the unrelated pLEM415 yields a measurable but moderate number of transformants. Numbers above bars represent the average number of transformants. **B.** Beta-glucuronidase assay with *Lactobacillus jensenii* pTRK892 transformants showed blue coloration after addition of X-GLU substrate. **C.** The optimized electroporation procedure is efficient across unrelated strains of *L. jensenii*. Shown is a comparison of transformation efficiencies obtained with the Luchansky [22] and optimized procedure with different clinical isolates. All electroporations were carried with 1.0 μg of DNA, 3X SMEB, 2% glycine, in 0.2 cm cuvettes and electric parameters as follows: voltage: 12.5 kV/cm, resistance: 400 Ω, capacitance: 25 μF. 3 biological replicates of 3 technical replicates were performed on different days (data are shown in S2 Table). Technical replicates are shown in the same color. (Wilcoxon signed-rank test p-value, ns p > 0.05, * p < = 0.05, ** p < = 0.01, *** p < = 0.001 and **** p < = 0.0001).

## Discussion

In this work, we optimized the transformation of *Lactobacillus jensenii*, a natural resident of the vaginal microflora and promising candidate chassis for vaginal microbiome engineering. We started with the protocol originally developed by Luchansky *et al.* [16] for the Lactobacillus group to which *L. jensenii* belongs, which only yielded a few dozen of clones ($4.7 \cdot 10^1$). After optimization of both competent preparation and electroporation conditions, we could obtain several thousands of transformants ($3.5 \cdot 10^3$ CFUs/μg plasmid on average with pTRK892). This improved efficiency will enable the construction and further screening of genetic libraries. Such libraries can be used for relatively high-throughput genetic engineering approaches, such as the establishment of collections of genetic elements involved in the regulation of gene expression [18], or the screening of combinatorial gene circuits [19]. Although the optimized procedure improves transformation efficiencies across unrelated strains of *L. jensenii*, additional fine-tuning would be necessary to maximize the absolute number of transformants obtained for any particular strain. Nevertheless, the transformation of multiple clinical isolates suggests the possibility of a personalized medicine approach for engineering microbiome therapeutics, with increased chances of successful colonization and a better resilience of the therapeutic strain in the natural flora environment.

In this work, we followed a simple greedy optimization procedure, wherein parameters are optimized sequentially using the best values identified in prior iterations. More refined Design of Experiment approaches [20], which are less susceptible to converging toward local optima, could be explored in future studies to access even higher efficiencies. A major factor hindering transformation that we have not explored here is the presence of restriction-modification systems in recipient bacteria. Several approaches—such as the heterologous expression of the recipient's putative methylases in the plasmid-producing strain or the *in vitro* methylation of purified plasmids using a range of commercially available enzymes—could be attempted to further improve our protocol [6].

Throughout this study, we observed some variation in transformation efficiencies across replicates. Some variability is expected in transformation efficiency within and mostly between different batches due to differences in growth, maintenance of the cold chain during cell preparation, storage and time before utilization. *L. jensenii* is microaerophilic and facultatively anaerobic. All experiments reported in this work were performed in GasPak anaerobic jars (5% $CO_2$), as we found these conditions to dramatically improve cell growth and competence. We initially attempted to culture cells in hermetic boxes with candles to create microaerophilic conditions (ca. 1–3% $CO_2$) (**S2 Fig**). While these conditions led to poorer performances, they nonetheless represent a workable alternative to the expensive GasPak system. Such an alternative might not be transposable to stricter anaerobes. Best performances were obtained with freshly prepared cells. We observed a dramatic drop in efficiency after 4 months of storage at -80°C, but only noticed a marginal decrease after 3 months. We thus recommend that cells are stored for no longer than 3 months. In these conditions, our protocol proved robust on different batches, storage conditions, consistently yielding at least a thousand transformants.

We showed that the two related plasmids pTRKH2 and pTRK892 (both developed by the Klaenhammer lab) provide good transformation efficiencies, with the latter yielding slightly more transformants (ca. 3500 vs ca. 2000). These data demonstrate that the regulatory elements driving the expression of *Erm(B)* (pTRKH2) and *Erm(C)* (pTRK892), as well as the *ppgm* promoter [21–23] and the ribosome binding site associated with the *gusA* reporter gene are all functional in *L. jensenii*. These elements can readily be used to express proteins of interest in future studies. The observed difference in efficiencies might be linked to differential effective activities of erythromycin resistance proteins produced from the two plasmids. Indeed, the promoter of Erm(B) is known to be positively regulated by erythromycin and might be weaker than Erm(C). The replication origin and mechanism might also play an important role, pTRKH2 carrying the pAMB1 rolling circle replication origin and pTRK892 carrying the pWVO1 theta replication origin. To the best of our knowledge, pLEM415-ldhl-mRFP1 is the most frequently used plasmid reported as functional in *L. jensenii* [24]. Interestingly, it could only be transformed with moderate efficiency in our hands. Since it is approximately the same size as other plasmids tested in this work, this low performance could be attributed to a poorly functional replication origin, which is more difficult to assess as it has not been well described [25].

While our work provides a first step towards enabling routine engineering of *L. jensenii*, many additional tools must be developed. For example, parts collections to control gene expression (such as promoters and RBS libraries with different strengths) will be needed to engineer genetic circuits operating reliably and with high precision [26, 27]. Chromosomal integration toolkits are also necessary to ensure stable genetic circuit maintenance without antibiotic selective pressure [28]. Finally, biocontainment, ethical and societal issues will need to be addressed before the use of stably engrafted engineered microbes becomes widespread. By enabling efficient plasmid transformation, our work provides a means to tackle these and other challenges in *Lactobacillus jensenii*.

## Supporting information

**S1 Fig. Map of the different vectors used in this study. A.** pTRKH2. **B.** pTRK892. **C.** pLEM415-ldhl-mRFP1.
(TIFF)

**S2 Fig. Use of hermetic jar and candles to create microaerophilic conditions suitable for *L. jensenii* culture. A.** Small plates (60mm diameter) are placed in a hermetically closed jar with

a burning candle. **B.** 100mm plates incubated in an airtight-closing bowl with candles. (TIFF)

**S1 Table. Primers sequences for colonies PCR of pTRKH2 *L. jensenii* transformants.** (DOCX)

**S2 Table. Raw data of CFUs transformations results.** (DOCX)

**S3 Table. Wilcoxon statistical analysis.** (DOCX)

**S1 Protocol. Preparation of electrocompetent *Lactobacillus jensenii* cells and electrotransformation.** (DOCX)

## Acknowledgments

We thank Pascale Serror (Micalis, Jouy-en-Josas, France) for insightful advice on lactobacilli. We thank members of our research groups and the CBS for fruitful discussions. We also thank the iGEM team Montpellier 2018, whose Vagineering project originally motivated this study. The team was composed of Léa Meneu, Elsa Fristot, Tamara Yehouessi, Younes Babaaziz, Marie Peras, Julien Mathieu, Leo Carrillo, Miguel Camacho Rufino, and Maxime Heintzé.

## Author Contributions

**Conceptualization:** Elsa Fristot, Jérôme Bonnet, Guillaume Cambray.

**Data curation:** Elsa Fristot, Jérôme Bonnet, Guillaume Cambray.

**Formal analysis:** Elsa Fristot, Guillaume Cambray.

**Funding acquisition:** Jérôme Bonnet.

**Investigation:** Elsa Fristot, Thomas Bessede, Miguel Camacho Rufino.

**Methodology:** Elsa Fristot, Hung-Ju Chang, Ana Zuniga, Jérôme Bonnet, Guillaume Cambray.

**Project administration:** Pauline Mayonove, Jérôme Bonnet, Guillaume Cambray.

**Resources:** Anne-Laure Michon, Sylvain Godreuil, Jérôme Bonnet, Guillaume Cambray.

**Supervision:** Jérôme Bonnet, Guillaume Cambray.

**Validation:** Jérôme Bonnet, Guillaume Cambray.

**Visualization:** Elsa Fristot, Jérôme Bonnet, Guillaume Cambray.

**Writing – original draft:** Elsa Fristot, Jérôme Bonnet, Guillaume Cambray.

**Writing – review & editing:** Elsa Fristot, Jérôme Bonnet, Guillaume Cambray.

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
