## [Decision Letter · Decision Letter 0]

2 Nov 2021

PONE-D-21-24476An Optimized Electrotransformation Protocol For Lactobacillus Jensenii

PLOS ONE

Dear Dr. Cambray,

Thank you for submitting your manuscript to PLOS ONE. After careful consideration, we feel that it has merit but does not fully meet PLOS ONE’s publication criteria as it currently stands. Therefore, we invite you to submit a revised version of the manuscript that addresses the points raised during the review process.

See below for some additional editorial remarks.

We look forward to receiving your revised manuscript.

Kind regards,

Sylvia Maria Bruisten, Ph.D

Academic Editor

PLOS ONE

Additional Editor Comments:

The reviewers have given useful comments that should be answered. One of them remarks that the study is not very innovative. Although this may be the case, that is not a criterium for rejection for PlosOne. The suggestion of another reviewer is to show electrotransformation results for more than one L jensenii strain. If this is possible to do it would certainly strenghten the results. If you would need more time for rebuttal in that case, please let me know.

Reviewers' comments:

Reviewer's Responses to Questions

**Comments to the Author**

1. Is the manuscript technically sound, and do the data support the conclusions?

Reviewer #1: Yes

Reviewer #2: No

Reviewer #3: Partly

2. Has the statistical analysis been performed appropriately and rigorously? 

Reviewer #1: Yes

Reviewer #2: No

Reviewer #3: Yes

3. Have the authors made all data underlying the findings in their manuscript fully available?

Reviewer #1: Yes

Reviewer #2: Yes

Reviewer #3: Yes

4. Is the manuscript presented in an intelligible fashion and written in standard English?

Reviewer #1: Yes

Reviewer #2: Yes

Reviewer #3: Yes

5. Review Comments to the Author

Reviewer #1: Review PLOS ONE An Optimized Electrotransformation Protocol For Lactobacillus Jensenii

This is well-conducted research and the paper is, for the most part, clearly written. I recommend publication with very minor corrections.

One thing that must be corrected is a paragraph in the Discussion where the presence of the ermB and ermC genes in the vectors is discussed (page 14). Two sentences in the paragraph are inconsistent with each other concerning which gene is on pTRKH2 and which gene is on the other vector being discussed.

Minor issues:

Key words: Why are ‘Acid Bacteria’ capitalized?

Abstract: Line 9 should ‘strain’ be ‘species’?

Abstract: Line 12 ‘cuvettes type’ should be ‘cuvette type’

Introduction (page 3): ‘bio engineered’ should be ‘bioengineered’

Introduction (last paragraph, six lines from end): ‘species optimized’ should be ‘species and optimized’

Figure 1B ‘additivs’ should be ‘additives’ and ‘vectors selection’ should be ‘vector selection’

Figure 1C why is ‘Cancer’ capitalized?

Figure 1 legend: ‘therapeutics’ should be ‘therapeutic’

Materials and media: third line in first paragraph: ‘Systems) ,’ remove unneeded blank space after the )

Materials and media: line 5 in first paragraph: why is Peptone capitalized?

Materials and media: in the Competent Cells Preparation section, line 3: ‘mL MRS fresh MRS’ should be ’mL fresh MRS’

Results: line 7 in first paragraph: ‘Addgene.’ Delete the period.

The word ‘Greedy’ is used twice in the paper but do you mean ‘Speedy’?

Table 1 legend: ‘abbreviation :’ should be ‘abbreviation:’ without the extra blank space

Table 1 ‘unknow’ should be ‘unknown’

Discussion: Should ‘Bacteroides thetaiotamicron’ be in italics?

Supplemental Figure 1 legend: panel D is mislabeled as a second panel C in the text.

Supplemental Figure 2 legend: Should the last sentence end with a period?

Supplemental Table 1: Should ‘primers’ be ‘primer’?

Supplemental Description of Electroporation protocol:

Step 1: Should ‘threw’ be ‘thaw’?

Step 4: Should ‘a tissues’ be ‘a tissue’? Should ‘electropore’ be ‘electroporate’?

The Time Constant Parameter: here you recommend 9-10 ms but the manuscript says 8-10.5 ms “(electroporation time constant must be in 8-10.5 ms range to show good efficiency)”

Step 8 mentions glass beads, but what size beads and is their use required? I don’t recall seeing them mentioned earlier in the paper.

Reviewer #2: The Manuscript by Fristot et al. describes the optimization of a protocol for the electrotransformation of Lactobacillus jensenii ATCC 25258.

The Manuscript is easily readable even though an excessive part of the text is dedicated to the introduction and a to a detailed description of standard methods.

The work appears as the simple setting of the condition for transforming this specific strain, just a preliminary step for any subsequent genetic research.

The title itself “An Optimized Electrotransformation Protocol For Lactobacillus Jensenii“ is misleading, because a single strain of the species has been tested and, considering the variability commonly observed among different strains of the same species, it is hard to assert that the described protocol may be used for the whole L.jensenii species.

I cannot see the originality of the work, as none of the materials have been generated during this study (both the host and the plasmids have been purchased from public sources) and the setup of the protocol has followed the strategies already described Luchansky and by Berthier in the cited references.

Even the transformation efficiency obtained is not better than that reported by the cited authors respectively for L. acidophilus and L.sakei (but in those papers multiple strain for each species were tested).

So, we think that the content of the manuscript isn’t innovative enough to be of interest for the readers.

Reviewer #3: The study is not novel but still provides some useful information. However, there are many problems that need to be addressed.

Major comments:

1. The "Introduction" is basically too long and didn't focus on the topic that the authors are talking about. The status of general plasmids/transformation in lactobacilli should be briefly discussed and mention the difficulty in L. jansenii.

2. the transformation can be strain-dependent, one strain is anyhow unconvincing and can not conclude an optimized electroporation for L. jansenii. The authors should try several different L. jansenii strains to validate the method.

3. the authors are highly recommended to try another replicon, pSH71 based plasmids, such as pNZ8148.

Minor comments:

P3: "In turn, the characterization...of these gram-positive bacteria". The authors should track the recent publications. One of the major factors that influence the transformation efficiency is the restriction-modification system, please refer to Zuo et al., 2019 doi.org/10.1021/acssynbio.9b00114, and Zuo et al., 2020 doi.org/10.1016/j.biotechadv.2020.107654.

P3: "Hervana bioengineered a strain..." a ref or link should be provided.

P5: ref for the L. jensenii strain was not clear.

P6: the description of buffer for competent cells preparation was not clear. Which was 3x and which was 1X?

P7: "according to the KAPA protocol of the PCR colony from Biosystem"? It was not clear.

P13: table 1: it is strange that pTRKH2 and pTRKH3-slp-GFP share the same replicon and antibiotic-resistant gene, but with different copy numbers and transformation outcomes. The authors should carefully check the difference between these two plasmids and interpret the results.

P14: the example of Bacteroides is not suitable here since you are talking about lactobacilli. Please refer to Bober et al., 2018 doi.org/10.1146/annurev-bioeng-062117-121019 and Zuo et al., 2020 doi.org/10.1016/j.biotechadv.2020.107654. Try to well organize the whole paragraph.

6. PLOS authors have the option to publish the peer review history of their article (what does this mean?). If published, this will include your full peer review and any attached files.

Reviewer #1: No

Reviewer #2: No

Reviewer #3: No

---

## [Author Response · Author response to Decision Letter 0]

10 Nov 2022

Reviewer #1: 

This is well-conducted research and the paper is, for the most part, clearly written. I recommend publication with very minor corrections.

We thank the reviewer for his positive assessment. We have corrected all minor points listed below in the revised version of the manuscript.

One thing that must be corrected is a paragraph in the Discussion where the presence of the ermB and ermC genes in the vectors is discussed (page 14). Two sentences in the paragraph are inconsistent with each other concerning which gene is on pTRKH2 and which gene is on the other vector being discussed.

Minor issues:

Key words: Why are ‘Acid Bacteria’ capitalized?

Abstract: Line 9 should ‘strain’ be ‘species’?

Abstract: Line 12 ‘cuvettes type’ should be ‘cuvette type’

Introduction (page 3): ‘bio engineered’ should be ‘bioengineered’

Introduction (last paragraph, six lines from end): ‘species optimized’ should be ‘species and optimized’

Figure 1B ‘additivs’ should be ‘additives’ and ‘vectors selection’ should be ‘vector selection’

Figure 1C why is ‘Cancer’ capitalized?

Figure 1 legend: ‘therapeutics’ should be ‘therapeutic’

Materials and media: third line in first paragraph: ‘Systems) ,’ remove unneeded blank space after the )

Materials and media: line 5 in first paragraph: why is Peptone capitalized?

Materials and media: in the Competent Cells Preparation section, line 3: ‘mL MRS fresh MRS’ 

should be ’mL fresh MRS’

Results: line 7 in first paragraph: ‘Addgene.’ Delete the period.

The word ‘Greedy’ is used twice in the paper but do you mean ‘Speedy’?

Table 1 legend: ‘abbreviation :’ should be ‘abbreviation:’ without the extra blank space

Table 1 ‘unknow’ should be ‘unknown’

Discussion: Should ‘Bacteroides thetaiotamicron’ be in italics?

Supplemental Figure 1 legend: panel D is mislabeled as a second panel C in the text.

Supplemental Figure 2 legend: Should the last sentence end with a period?

Supplemental Table 1: Should ‘primers’ be ‘primer’?

Supplemental Description of Electroporation protocol:

Step 1: Should ‘threw’ be ‘thaw’?

Step 4: Should ‘a tissues’ be ‘a tissue’? Should ‘electropore’ be ‘electroporate’?

The Time Constant Parameter: here you recommend 9-10 ms but the manuscript says 8-10.5 ms “(electroporation time constant must be in 8-10.5 ms range to show good efficiency)”

Step 8 mentions glass beads, but what size beads and is their use required? I don’t recall seeing them mentioned earlier in the paper.

Reviewer #2:

The title itself “An Optimized Electrotransformation Protocol For Lactobacillus Jensenii“ is misleading, because a single strain of the species has been tested and, considering the variability commonly observed among different strains of the same species, it is hard to assert that the described protocol may be used for the whole L.jensenii species.

This is indeed a common problem and we thank the reviewer for this comment. The same issue is also raised by reviewer 3. To answer this, we have obtained 3 independent L. jensenii from a bank established in a local hospital (Arnaud de Villeneuve UMR IRD224-CNRS5290-UM). We tested our protocol on these additional strains and added the results in Figure 4. We found that the protocol does permit to increase the transformability of all strain tested relative to the original procedure, though to various extent (10 to 40-fold increase).

I cannot see the originality of the work, as none of the materials have been generated during this study (both the host and the plasmids have been purchased from public sources) and the setup of the protocol has followed the strategies already described Luchansky and by Berthier in the cited references.

We must agree that this is not a tremendously original work. However, we did combined parameters identified from different sources (Luchansky et al., 1988; Luchansky et al., 1989; Chang et al., 1989; Berthier et al., 1996). In doing so, we believe that we contributed some clarity as to what procedure can be followed for other endeavors of this type. We also think our results on L. jensenii can be valuable to others.

Even the transformation efficiency obtained is not better than that reported by the cited authors respectively for L. acidophilus and L.sakei (but in those papers multiple strain for each species were tested).

Transformation efficiency can vary tremendously between strains and species. Some species transform well using standardized procedures, other are well-studied and suitable protocols have already been published. We found L. Jensenii to be particularly difficult to transform and experienced more difficulties than expected in optimizing the efficiency reported in this manuscript. That we identified a protocol yielding effixiencies on par with more amenable species is no small feat in our opinion. We very much hope that our data will promote the use of L. Jensinii to complement other more amenable strain from the vaginal microflora such as L. crispatus or L. gasseri.

So, we think that the content of the manuscript isn’t innovative enough to be of interest for the readers.

Our work combined optimization parameters identified from multiple source and report transformability on par with more commonly used species. This can serve as a foundation for other endeavors of that type and enable the genetic manipulation of the most widely distributed species across vaginal microflora worldwide.

Reviewer #3: The study is not novel but still provides some useful information. However, there are many problems that need to be addressed.

Major comments:

1. The "Introduction" is basically too long and didn't focus on the topic that the authors are talking about. The status of general plasmids/transformation in lactobacilli should be briefly discussed and mention the difficulty in L. jansenii.

We certainly agree with the reviewer and we apologize for that. We have now completely revised the introduction to make it much shorter and to the point.

2. the transformation can be strain-dependent, one strain is anyhow unconvincing and can not conclude an optimized electroporation for L. jansenii. The authors should try several different L. jansenii strains to validate the method.

The same issue is raised by reviewer 2. To answer this, we have obtained 3 independent L. jensenii from a bank established in a local hospital (Arnaud de Villeneuve UMR IRD224-CNRS5290-UM). We tested our protocol on these additional strains and added the results in Figure 4. We found that the protocol does permit to increase the transformability of all strain tested relative to the original procedure, though to various extent (10 to 40 fold increase).

3. the authors are highly recommended to try another replicon, pSH71 based plasmids, such as pNZ8148.

We would like to thank the reviewer for this suggestion. In fact, we used vector pTRK892 which precisely carries a pSH71 origin of replication. pSH71 is itself a derivative of pWVO1, a low copy origin of replication that is functional in both gram negative and positive bacteria. We have amended our manuscript to clarify all this.

Minor comments:

P3: "In turn, the characterization...of these gram-positive bacteria". The authors should track the recent publications. One of the major factors that influence the transformation efficiency is the restriction-modification system, please refer to Zuo et al., 2019doi.org/10.1021/acssynbio.9b00114, and Zuo et al., 2020 doi.org/10.1016/j.biotechadv.2020.107654

We have added a sentence in the conclusion to mention this as a perspective. We thank the reviewer for pointing out these interesting papers and have included them as references in the revised manuscript.

P13: table 1: it is strange that pTRKH2 and pTRKH3-slp-GFP share the same replicon and antibiotic-resistant gene, but with different copy numbers and transformation outcomes. The authors should carefully check the difference between these two plasmids and interpret the results.

We thank the reviewer for highlighting this discrepancy, which prompted us to fully sequence these two plasmids. Our result show that the replication origin of our version of pTRKH3-slp-GFP contains a deletion. We have contacted Addgene to report this problem. We have now removed pTRKH3-slp-GFP from our manuscript. As it was a direct derivative of pTRKH2, which is used throughout the manuscript, we think the loss of information due to this redaction is minimal.

---

## [Decision Letter · Decision Letter 1]

25 Nov 2022

PONE-D-21-24476R1

An Optimized Electrotransformation Protocol for Lactobacillus Jensenii

PLOS ONE

Dear Dr. Cambray,

Thank you for submitting your manuscript to PLOS ONE. After careful consideration, we feel that it has merit but does not fully meet PLOS ONE’s publication criteria as it currently stands. Therefore, we invite you to submit a revised version of the manuscript that addresses the points raised during the review process.

Two of the three reviewers agree with this manuscript in its present form. The third reviewer has still some usefull comments to further improve the manuscript.

We look forward to receiving your revised manuscript.

Kind regards,

Sylvia Maria Bruisten, Ph.D

Academic Editor

PLOS ONE

Journal Requirements:

Reviewers' comments:

Reviewer's Responses to Questions

**Comments to the Author**

1. If the authors have adequately addressed your comments raised in a previous round of review and you feel that this manuscript is now acceptable for publication, you may indicate that here to bypass the “Comments to the Author” section, enter your conflict of interest statement in the “Confidential to Editor” section, and submit your "Accept" recommendation.

Reviewer #1: All comments have been addressed

Reviewer #2: (No Response)

Reviewer #3: All comments have been addressed

2. Is the manuscript technically sound, and do the data support the conclusions?

Reviewer #1: Yes

Reviewer #2: Yes

Reviewer #3: Yes

3. Has the statistical analysis been performed appropriately and rigorously? 

Reviewer #1: Yes

Reviewer #2: No

Reviewer #3: Yes

4. Have the authors made all data underlying the findings in their manuscript fully available?

Reviewer #1: Yes

Reviewer #2: No

Reviewer #3: Yes

5. Is the manuscript presented in an intelligible fashion and written in standard English?

Reviewer #1: Yes

Reviewer #2: Yes

Reviewer #3: Yes

6. Review Comments to the Author

Reviewer #1: (No Response)

Reviewer #2: Major comments

Most of the previous comments has been properly addressed by the authors and we noticed the inclusion of three new strains of L. jensenii in this study.

Nevertheless, the results are improperly reported and thus the conclusions are not appropriate. The authors must correct two main faults:

1) Data are shown only by graphs. The original numerical data and their averages should be reported, at least as supplementary information.

2) Trials have been performed in biological and technical replicates, but no statistical analysis is reported (neither in methods nor in results). Therefore, no statement about the significance of the observations is acceptable.

Minor comments

Please check these possible mistakes (Raws are referred to the pdf version, disregarding blank lines):

P2 raws 5-6

low abundances = low abundance?

P2 raw 10

capacities = capacity?

P3 raw 11

to secreting = to secrete?

P3 raw 18

medium sized = medium-sized?

P4 raw 16

Cultures were incubating = Cultures were incubated?

P4 raws 22-23

pTRKH2 (kind gift from Michela Lizier, Addgene plasmid # 27168) = This reference is that of pTRKH3slpGFP from the previous version . pTRKH2 is not from Lizier’s lab

P5 raw 18

is as follow = is as follows?

P10 raw 2

different DNA quantity = different DNA quantities?

P11 raw 20

a few dozens = a few dozen?

P12 raw 4

any particular strains = any particular strain?

P12 raw 15

improve on our protocol = improve our protocol?

P13 raw 23

with high-precision = with high precision?

P14 raw 2

provides a mean = provides a means

p 16 S1 table

The whole 16S gene is shorter than the size indicated for the amplicon obtained with primers pFOR 16S L. jensenii and pREV 16S L. jensenii – check the value

Reviewer #3: (No Response)

7. PLOS authors have the option to publish the peer review history of their article (what does this mean?). If published, this will include your full peer review and any attached files.

Reviewer #1: No

Reviewer #2: No

Reviewer #3: No

---

## [Author Response · Author response to Decision Letter 1]

8 Jan 2023

REviewer #2 : Raw datas are now fully available in table 2 from supplementary data. We added a Wilcoxon analysis on our graphs to support our observations in the results section. P-values data are available in table 3 from supplementary data. We also thank the reviewer for highlighting an inconsistancy regarding the size of the amplicon in our supplementary table 1, we rechecked the sequence between our two primers and indeed the correct amplicon size is well 1276bp.

---

## [Editor Report · Decision Letter 2]

12 Jan 2023

An Optimized Electrotransformation Protocol for Lactobacillus Jensenii

PONE-D-21-24476R2

Dear Dr. Cambray,

We’re pleased to inform you that your manuscript has been judged scientifically suitable for publication and will be formally accepted for publication once it meets all outstanding technical requirements.

Kind regards,

Sylvia Maria Bruisten, Ph.D

Academic Editor

PLOS ONE

Additional Editor Comments (optional):

All comments have now been addressed to satisfaction
---

## [Editor Report · Acceptance letter]

10 Feb 2023

PONE-D-21-24476R2 

An optimized electrotransformation protocol for *Lactobacillus jensenii*

Dear Dr. Cambray:

I'm pleased to inform you that your manuscript has been deemed suitable for publication in PLOS ONE. Congratulations! Your manuscript is now with our production department. 

Kind regards, 

on behalf of

Dr. Sylvia Maria Bruisten 

Academic Editor

PLOS ONE